# Enterotoxin- and Antibiotic-Resistance-Encoding Genes Are Present in Both Coagulase-Positive and Coagulase-Negative Foodborne *Staphylococcus* Strains

Acácio Salamandane [1,2,*], Jessica Oliveira [1], Miguel Coelho [1], Beatriz Ramos [3,4], Mónica V. Cunha [3,4], Manuel Malfeito-Ferreira [1] and Luisa Brito [1]

1   LEAF—Linking Landscape, Environment, Agriculture and Food Research Centre, Associated Laboratory TERRA, Instituto Superior de Agronomia, Universidade de Lisboa, Tapada da Ajuda, 1349-017 Lisboa, Portugal; jessicasemedo97@gmail.com (J.O.); migueljgcoelho2009@hotmail.com (M.C.); mmalfeito@isa.ulisboa.pt (M.M.-F.); lbrito@isa.ulisboa.pt (L.B.)
2   Faculdade de Ciências de Saúde, Universidade Lúrio, Nampula 4250, Mozambique
3   Centre for Ecology, Evolution and Environmental Changes (cE3c) & CHANGE—Global Change and Sustainability Institute, Faculdade de Ciências, Universidade de Lisboa, 1749-016 Lisboa, Portugal; beatriz.g.ramos@gmail.com (B.R.); mscunha@fc.ul.pt (M.V.C.)
4   Biosystems and Integrative Sciences Institute (BioISI), Faculdade de Ciências, Universidade de Lisboa, 1749-016 Lisboa, Portugal
*   Correspondence: salamandane@gmail.com

**Abstract:** Food poisoning by staphylococcal enterotoxins (SE) is a major cause of foodborne illness, often associated with coagulase-positive staphylococci (CPS). The increase in the number of methicillin-resistant *Staphylococcus aureus* (MRSA) strains is another major problem associated with CPS. However, reports of the association of SE and methicillin-resistant *Staphylococcus* with coagulase-negative staphylococci (CNS) are beginning to re-emerge. In this context, the aim of this study is to investigate the presence of staphylococcal enterotoxin genes and to characterize the phenotypic and genotypic antimicrobial resistance in 66 isolates of *Staphylococcus* spp. (47 CNS and 19 CPS) recovered from ready-to-eat (RTE) street food sold in Maputo, Mozambique. Seven virulence genes encoding SE (*sea*, *seb*, *sec*, *sed* and *see*) and two toxins (*hlb* and *sak*) were screened by multiplex PCR (MPCR). Antimicrobial resistance against 12 antibiotics was evaluated by the disk diffusion method. The presence of genes encoding resistance to penicillin, methicillin, vancomycin and erythromycin (*blaZ*, *mecA*, *vancA*, *vancB*, *ermA*, *ermB* and *ermC*) were also screened by PCR. At least one of the seven virulence genes assessed in this study was detected in 57.9% and 51% of CPS and CNS isolates, respectively. In CPS isolates, the most frequent gene was *hlb* (47.4%), followed by *sec* (15.8%) and *sea*, *seb* and *sed* genes with 5.3% each. In CNS isolates, the most frequent gene was *sec* (36.2%) followed by *sak* (17%), *hlb* (14.9%), *sed* (12.8%) and *seb* (6.4%). Five of the twelve CPS in which virulence genes were detected were also antibiotic-resistant. All the CNS isolates harboring virulence genes (*n* = 27, 57.4%) were antimicrobial-resistant. The prevalence of multidrug resistance was higher (59.6%) in CNS than in CPS (26.3%) isolates. Regarding the presence of antibiotic-resistance genes, *blaZ* (penicillin-resistant) was the most frequent in both CPS (42.1%) and CNS (87.2%), followed by the *mecA* (encoding methicillin resistance) and *vancA* genes (vancomycin-resistant), which represented 36.8% and 31.6% in CPS isolates and 46.8% in CNS isolates, respectively. The prevalence of vancomycin-resistant staphylococci has been increasing worldwide and, to our knowledge, this is the first study to report the occurrence of vancomycin-resistant staphylococci in Mozambique. These results emphasize the need to investigate CNS isolates in parallel with CPS, as both constitute public health hazards, given their potential to produce SE and spread antimicrobial resistance genes.

**Keywords:** Maputo; Mozambique; coagulase-negative staphylococci (CNS); staphylococcal enterotoxins (SE); vancomycin resistance; methicillin resistance; multidrug resistance

## 1. Introduction

*Staphylococcus* species are frequently found on the human skin and nasal mucosa, but can also be found in soil, water and food products, among others [1–4]. The contamination of the food matrix by *Staphylococcus* spp. can occur via contaminated raw materials, such as meat and non-pasteurized milk. Additionally, cross-contamination from utensils and equipment as well as from other contaminated products, particularly those subjected to handling [5,6], are also relevant transmission routes. The pathogenicity of *S. aureus* is related to the presence of several virulence factors, including *S. aureus* enterotoxins (SE), hemolysins encoded by *hla* and *hlb* genes and staphylokinase encoded by the *sak* gene [6–9]. Hemolysins are involved in the tissue adhesion of the pathogens, colonization and tissue invasiveness, thus promoting pathogenicity [6,7]. Staphylokinase is a bacteriophage-encoded protein expressed by lysogenic strains of *S. aureus* [10]. It is a protein participating in the disintegration of fibrin and considered one of the *S. aureus* virulence factors [10,11].

Food poisoning by Staphylococcal enterotoxin is one of the most prevalent food-borne illnesses, often associated with coagulase-positive *Staphylococcus* (CPS) [2]. Currently, more than 23 staphylococcal or staphylococcal-related enterotoxins are known, and their ingestion in amounts ranging from 20 ng to 1 mg can cause severe symptoms in humans [6,12]. These proteins are resistant to proteolysis and are thermostable [13–15]. Although the precise mechanisms of action of SE are still unclear, it is known that food-borne SE directly affects the intestinal epithelium and the vagus nerve (pneumogastric nerve), causing the stimulation of the emetic center [16]. Classical enterotoxins A–E have been associated with more than 90% of staphylococcal food poisoning cases worldwide [6] and often linked with CPS [17,18]. However, reports of the association of enterotoxins from food poisoning with coagulase-negative staphylococcus (CNS) are beginning to re-emerge [17,19–22].

Antibiotic resistance, particularly methicillin-resistant *Staphylococcus aureus* (MRSA), is another problem associated with staphylococci [23,24]. In recent decades, the incidence of MRSA strains has increased worldwide [12], which has compromised the treatment of nosocomial infections, especially in low-income countries [25]. In 2019, an estimated 1.2 million deaths were directly related to antibiotic-resistant strains, and *S. aureus* was the second most important attributable or associated antibiotic-resistant species [26]. However, antibiotic resistance has also been reported in coagulase-negative staphylococcal strains [27,28]. Furthermore, several studies showed that CNS may be reservoirs for the *mecA* genes encoding penicillin-binding protein *(PBP2a)* [27–30], thus emphasizing the importance of CNS in public health.

When associated with poor hygiene practices by handlers and poor hygienic conditions of the exposure environment, ready-to-eat (RTE) street food presents a potential risk of transmission of staphylococcal strains [31–36]. In Maputo, Mozambique, street foods are sold in environments with poor sanitation conditions and lack of potable water, which increases the risk of street food contamination [33,34,37,38]. This study is a follow-up to previous research by Salamandane et al. (2021) [31], and aims to evaluate the antimicrobial resistance profile and the occurrence of both virulence- and antibiotic-resistance genes in 66 coagulase-positive and coagulase-negative *Staphylococcus* isolates previously recovered from RTE foods sold on the streets of Maputo, Mozambique.

## 2. Material and Methods

This study used 70 isolates of presumptive *Staphylococcus* previously recovered from 81 RTE street food sold in Maputo, described in Salamandane et al. (2021) [31]. In general, all the food samples were considered unsatisfactory for consumption [31]. *Staphylococcus* spp. presumptive colonies were obtained after incubation for 48 h at 37 °C onto Chromogenic Baird Parker Agar—CBPA (Biokar Diagnostics, Beauvais, France) according to ISO 6888 [39], as described in Salamandane et al. (2021) [31]. From each plate, one CFU that presented typical characteristics of *Staphylococcus* was selected.

## 2.1. Identification of Staphylococcus

Biochemical tests were carried out to characterize the presumptive *Staphylococcus* colonies from CBPA plates, namely Gram staining, catalase and oxidase tests [31]. Subsequently, coagulase tests were performed in Coagulase Rabbit Plasma (Biokar Diagnostics, Beauvais, France) and in Baird Parker RPF Agar (Biokar Diagnostics, Beauvais, France), according to ISO 6888-2/A1-2003 [40]. The presence of the coagulase gene was confirmed by PCR amplification targeting the *coa* gene (600–900 bp) with the coa Forward 5′ CGA-GACCAAGATTCAACAAG 3′ and coa Reverse 5′ AAAGAAAACCACTCACATCA 3′ primers, as previously described by Adame-Gómez et al. (2020) [2]. Molecular identification was performed targeting the 16S rRNA gene (1500 bp) with Bac27F forward primer (5-AGAGTTTGGATCMTGGCTCAG-3) and Univ1492R universal reverse primer (5-CGGTTACCTTGTTACGACTT-3). The amplified products were sequenced (STAB VIDA, Caparica, Portugal) and the resulting sequences were analyzed as follows.

## 2.2. Species Identification and Phylogenetic Analysis

The sequenced 16S rRNA amplicons were submitted to blastn using megablast algorithm against Reference RNA sequences database (RefSeq RNA) from the National Center of Biotechnology Information (NCBI) for identification. The reference 16S rRNA sequences of the twelve species identified by BLAST search and the outgroup (*Macrococcus caseolyticus*) were retrieved from NCBI RefSeq RNA database (Table S1). The multiple alignment of the reference 16S rRNA sequences and those of the 70 presumptive *Staphylococcus* isolates from this study was performed in MEGA X version 10.2.6 (PA, USA) using MUSCLE algorithm (v10.0.5). The alignment was truncated to achieve a core alignment of 1 kb. For phylogenetic analysis, the core alignment was used to construct a maximum-likelihood tree with 100 replicates, using General Time Reversible substitution model and Gamma distributed with invariant sites rate, in MEGA X (v10.0.5). Tree editing and annotation was performed using iTol (v6.5.4) [41].

## 2.3. PCR for the Identification of Virulence and Antibiotic Resistance Genes

The presence of seven genes encoding toxin hemolysin (*hlb*), staphylokinase (*sak*), and five enterotoxins (*sea*, *seb*, *sec*, *sed*, and *see*) was investigated. For DNA extraction, six bacterial colonies grown on TSA plates for 18 ± 2 h at 37 °C were suspended in 300 μL of sterile ultrapure water and incubated in a boiling bath for 10 min [42]. Subsequently, the tubes were centrifuged at 16,000× *g* for 12 min, and the lysate supernatants (lysates) were stored at −20 °C. The master mix for MPCR was prepared with 12.5 μL of Taq DNA Polymerase NZYTaq II2x Colourless Master Mix (MZYTech, Lisboa, Portugal), 1 μL of each forward and reverse primer indicated in Table 1 (final concentration 0.3 μM), 2 μL of template DNA and sterile ultrapure water to fill 25 μL of total volume. Based on the differences in the hybridization temperatures, thermocycling reactions were performed in two different MPCRs, one for the *hlb*, *sea* and *see* genes and the other for the *sak*, *seb*, *sec* and *sed* genes, according to Adame-Gómez et al. (2020) [2].

**Table 1.** Primers used for the detection of virulence genes and respective product sizes and hybridization conditions (Jarraud et al. (2002) [43] and Adame-Gómez et al. (2020) [2]).

| Virulence Factor (Target Gene) | | Primer Sequence (5′–3′) | Amplicon Size (bp) | Annealing Temperature (°C)/Time (s) |
|---|---|---|---|---|
| Enterotoxin A (*sea*) | seaF<br>seaR | TGCAGGGAACAGCTTTAGGC<br>GTGTACCACCCGCACATTGA | 250 | 52/30 |
| Enterotoxin B (*seb*) | sebF<br>sebR | ATTCTATTAAGGACACTAAGTTAGGG<br>ATCCCGTTTCATAAGGCGAGT | 400 | 52/45 |
| Enterotoxin C (*sec*) | secF<br>secR | GTAAAGTTACAGGTGGCAAAACTTG<br>CATATCATACCAAAAAGTATTGCCGT | 297 | 52/45 |

**Table 1.** *Cont.*

| Virulence Factor (Target Gene) | | Primer Sequence (5′–3′) | Amplicon Size (bp) | Annealing Temperature (°C)/Time (s) |
|---|---|---|---|---|
| Enterotoxin D (*sed*) | sedF<br>sedR | GAATTAAGTAGTACCGCGCTAAATAATATG<br>GCTGTATTTTTCCTCCGAGAGT | 492 | 52/45 |
| Enterotoxin E (*see*) | seeF<br>seeR | CAAAGAAATGCTTTAAGCAATCTTAGGC<br>CACCTTACCGCCCAAAGCTG | 480 | 52/30 |
| Hemolysin (*hlb*) | hlbF<br>hlbR | GTGCACTTACTGACAATAGTGC<br>GTTGATGAGTAGCTACCTTCAGT | 300 | 52/30 |
| Staphylokinase (*sak*) | sakF<br>sakR | ATCCCGTTTCATAAGGCGAGT<br>CACCTTACCGCCCAAAGCTG | 260 | 52/45 |

The screening for the antibiotic-resistance genes used seven primer pairs for the detection of genes encoding resistance to four antimicrobials (Table 2). The master mix for MPCR was prepared with 12.5 µL of Taq DNA Polymerase NZYTaq II2x Colourless Master Mix (MZTech, Lisboa, Portugal), 1 µL of each forward and reverse primer indicated in Table 2 (final concentration 0.3 µM), 2 µL of DNA template and sterile ultrapure water to complete 25 µL of the total volume. For *mecA* detection, the reaction mixtures were subjected to the following amplification conditions: initial denaturation, 94 °C for 5 min, followed by 30 cycles of denaturation 94 °C for 30 s, annealing 52 °C for 30 s, elongation 72 °C for 30 s, and a final elongation at 72 °C for 5 min. The detection of *blaZ* was performed with an initial denaturation at 94 °C for 5 min, followed by 30 cycles of denaturation 94 °C for 20 s, annealing 60 °C for 30 s, elongation 72 °C for 90 s, and a final elongation at 72 °C for 5 min. For the detection of erythromycin- (*ermA*, *ermB* and *ermC*) and vancomycin (*vancA* and *vancB*)-resistance genes, monoplex PCRs were performed with the pairs of primers indicated in Table 2, as described by Prunier et al. (2003) [44]. For the detection of vancomycin (*vancB*)-resistance genes, monoplex PCRs were performed as described by Al-Amery [45] (Table 2).

**Table 2.** Primers used for the detection of antimicrobial-resistance genes and the respective amplicon product sizes.

| Resistance to the Antibiotic | Primer for the Target Gene | Primer Sequence (5′–3′) | Amplicon Size (bp) | References |
|---|---|---|---|---|
| Penicillin | *blaZ-F*<br>*blaZ-R* | AAGAGATTTGCCTATGCTTC<br>GCTTGACCACTTTTATCAGC | 170 | [46,47] |
| Methicillin | *mecA-F*<br>*mecA-R* | TCCAGATTACAACTTCACCAGG<br>CCACTTCATATCTTGTAACG | 180 | [48] |
| Erythromycin | *ermA-F*<br>*ermA-R* | AAGCGGTAAACCCCTCTGA<br>TTCGCAAATCCCTTCTCAAC | 190 | [49] |
| | *ermB-F*<br>*ermB-R* | TCAAAACATAATATAGATAAA<br>GCTAATATTGTTTAAATCGTCAAT | 642 | [44] |
| | *ermC-F*<br>*ermC-R* | AATCGTCAATTCCTGCATGT<br>TAATCGTGGAATACGGGTTTG | 299 | [49] |
| Vancomycin | *vancA-F*<br>*vancA-R* | GGCAAGTCAGGTGAAGATG<br>ATCAAGCGGTCAATCAGTTC | 713 | [45] |
| | *vancB-F*<br>*vancB-R* | GTGACAAACCGGAGGCGAGGA<br>CCGCCATCCTCCTGCAAAAAA | 430 | |

All PCR reactions were run in a thermocycler GeneAmp® PCR System 9700, Applied Biosystems (Bio-Rad Laboratories, Segrate, Milan, Italy). The resulting PCR products were resolved on 2% (m/v) agarose gels in 1× TAE buffer, in a EC330 Thermo Fisher Scientific tank (Atlanta, GA, USA) at 8 V/cm for 60 min. The gels were stained with GelRed (Frilabo, Maia, Portugal) and analyzed using a Gel Doc™ EZ System (Bio-Rad Laboratories, Segrate, Milan, Italy). For calculating the size of the PCR products, the molecular marker 100 bp DNA Ladder (Invitrogen, CA, USA) was used.

### 2.4. Antimicrobial-Resistance Profile

The antimicrobial-resistance profile was evaluated for the 70 presumptive *Staphylococcus* spp. isolates using the disk diffusion method on Mueller–Hinton (MH) agar plates (Biokar Diagnostics, Beauvais, France) with antibiotic discs (Liofilchem, Roseto degli Abruzzi, Italy), according to the Clinical Laboratory Standards Institute (CLSI, 2021) [50]. Isolated colonies grown on Trypto-Casein-Soy agar (Biokar Diagnostics, Beauvais, France) for 22 ± 2 h at 37 °C were suspended in sterile saline until the turbidity was equivalent to the McFarland 0.5 standard (ca. $10^6$ CFU/mL). Of the resulting bacterial suspensions, 100µL were used to inoculate MH plates in the conditions described by CLSI [50]. After the deposition of the antibiotic discs, the plates were incubated for 18 ± 2 h at 37 °C. Twelve antibiotics were tested: cefoxitin (FOX) 30 µg; ampicillin (AMP) 10 µg; methicillin (MET) 5 µg; vancomycin (VAN) 5 µg; penicillin G (PEN) 10 µg; chloramphenicol (CHL) 30 µg; tetracycline (TET) 30 µg; gentamicin (GEN) 10 µg; trimethoprim/sulfamethoxazole (SXT) 23.75/1.25 µg; erythromycin (ERY) 16 µg; ciprofloxacin (CIP) 5 µg; and levofloxacin (LVX) 5 µg. In each 90 mm diameter plate, five different antibiotic discs were placed. For each isolate, two replicates were performed.

### 2.5. Data Interpretation

For the evaluation of the antimicrobial resistance profile, the inhibition halos were measured (millimeter) and compared to those described in the CLSI (2021) [50]. Isolates were considered non-susceptible to a given antibiotic when they showed intermediate or full resistance according to the CLSI clinical breakpoints. Multidrug resistance was considered as non-susceptibility to at least one agent in three or more antimicrobial categories and/or resistance to methicillin [51].

### 3. Results

The 70 presumptive staphylococcal isolates used in this study were identified based on 16S rRNA gene sequence analysis (Figure 1). *Staphylococcus aureus* represent 27% of the 70 isolates under study and correspond to all CPS isolates. The most representative CNS species are *S. warneri* (22.9%), *S. saprophyticus* (18.6%), *S. xylosus* (10%) and *S. pasteuri* (5.6%). Four isolates initially classified by biochemical methods as belonging to the *Staphylococcus* genus were identified as belonging to the species *Mammaliicoccus sciuri* (Figure 1). These four isolates did not show in vitro methicillin or erythromycin resistance. Two of the three *Mammaliicoccus sciuri* isolates presented virulence genes and one was resistant to vancomycin and to penicillin G.

### 3.1. Detection of Virulence Genes

Virulence genes were detected among the 39 identified *Staphylococcus* isolates (Figure 1). CPS (63.2%) and CNS (57.4%) showed at least one of the seven genes assessed in this study (Figure 1). The *hlb* gene was detected in 47.4% of the CPS and 14.9% of the CNS isolates. In CPS, *hlb* was the most frequent virulence gene, followed by *sec* 15.8% (Table 3). In the CNS, *sec* was the most frequent gene (36.2%), followed by *sak* (17%) and *hlb* (14.9%) (Table 3).

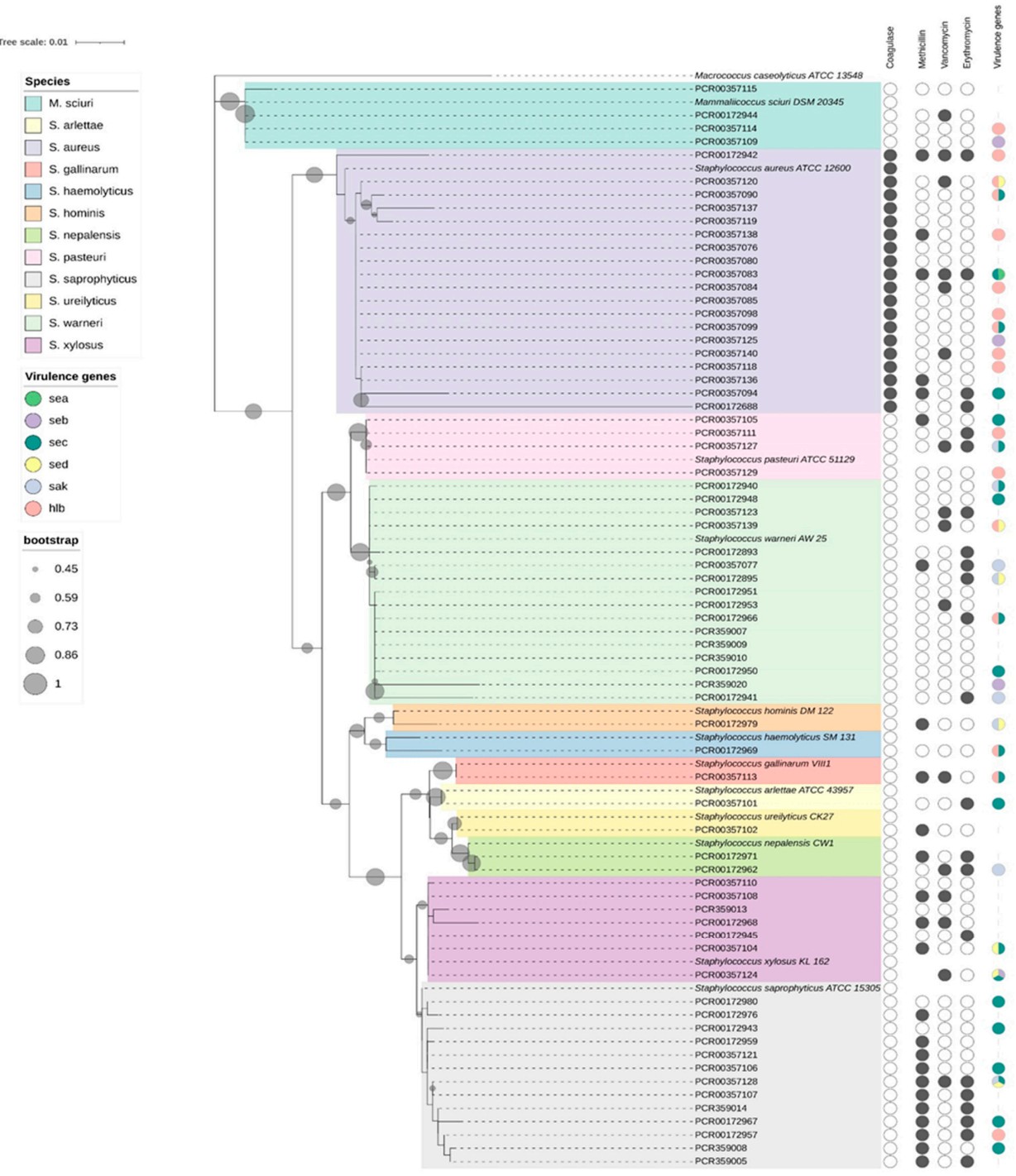

**Figure 1.** Maximum-likelihood tree of 16S rRNA sequences retrieved from the 70 presumptive *Staphylococcus* isolated from RTE street food. The tree was computed using the General Time Reversible (GTR) substitution model and Gamma distributed with invariant sites rate. Bootstrap values were based on 100 replicates. *Macrococcus caseolyticus* ATCC 13548 was used as the outgroup. Coagulase and antimicrobial-resistance traits are displayed as binary data (grey—presence; white—absence). The detection of the screened virulence genes is informed by colored circles. Data on the *see* gene are not presented in the tree since no isolate was found to harbor this gene. Tree editing and annotation was performed with iTol (v6.5.4).

**Table 3.** Percentage (%) of coagulase-positive Staphylococcus (CPS) and coagulase-negative Staphylococcus (CNS) isolates in which virulence genes were detected by PCR.

| Virulence Genes | CPS (*n* = 19) | CNS (*n* = 47) | Total (*n* = 66) |
|---|---|---|---|
| *sea* * | 1 | 0 | 1 |
| *seb* * | 1 | 3 (6.4%) | 4 (6.1%) |
| *sec* * | 3 (15.8%) | 17 (36.2%) | 20 (30.3%) |
| *sed* * | 2 | 6 (12.8%) | 8 (12.1%) |
| *see* * | 0 | 0 | 0 |
| *hlb* | 9 (47.4%) | 7 (14.9%) | 16 (24.2%) |
| *sak* | 0 | 8 (17%) | 8 (12.1%) |
| Total | 16 (84.2%) | 41 (87.2%) | 57 (86.4%) |

* Genes encoding staphylococcal enterotoxins (SE).

In the CPS, except for the *seb* gene that did not co-occur with the other screened genes, all genes co-occurred with *hlb* (*hlb* and *sea* in one isolate; *hlb* and *sec*, and *hlb* and *sed* in two isolates, each) (Figure 1). In the CNS, the coexistence of *hlb* and *seb*, and *hlb* and *sec* was detected in two isolates, respectively: *sed* and *sak* in one isolate, and the co-occurrence of *seb*, *sec* and *sed* genes in one isolate (Figure 1).

The *see* gene was not detected in any of the isolates. The *sak* gene was detected exclusively in CNS isolates (8/47) (Table 3).

### 3.2. Antimicrobial-Resistance Profile

The antimicrobial-resistance profiles of the isolates evaluated in this study showed a high occurrence of resistance to β-lactam antibiotics and a low occurrence of resistance to non-β-lactam antibiotics (Table 4). The highest resistances were to β-lactams PEN (83.3%), FOX and MET (37.9% each) and VAN (27.7%) (Table 4). Among non-β-lactams, the highest resistance rate was against ERY (33.3%) followed by TET and SXT (6.1% each).

**Table 4.** Antimicrobial-resistance profiles of the 66 Staphylococcus spp. isolates from RTE street food.

| Antibiotic Group | Antibiotic | CPS (*n* = 19) | CNS (*n* = 47) | Total (*n* = 66) |
|---|---|---|---|---|
| β-lactam | FOX (30 µg) | 5 (26.3%) | 20 (42.6%) | 25 (37.9%) |
| | AMP (10 µg) | 5 (26.3%) | 6 (12.8%) | 11 (16.7%) |
| | MET (5 µg) | 5 (26.3%) | 20 (42.6%) | 25 (37.9%) |
| | VAN (5 µg) | 5 (26.3%) | 10 (21.3%) | 15 (27.7%) |
| | PEN (10 µg) | 11 (57.9%) | 44 (93.6%) | 55 (83.3%) |
| Non- β-lactam | CHL (30 µg) | 0 | 2 (4.3%) | 2 (3%) |
| | TET (30 µg) | 1 | 3 (6.4%) | 4 (6.1%) |
| | GEN (10 µg) | 0 | 2 (4.3%) | 2 (3%) |
| | SXT (23.75/1.25 µg) | 0 | 4 (8.5%) | 4 (6.1%) |
| | ERY (16 µg) | 4 (21.1%) | 18 (38.3%) | 22 (33.3%) |
| | CIP (5 µg) | 0 | 0 | 0 |
| | LVX (5 µg) | 0 | 0 | 0 |

Cefoxitin (FOX), Ampicillin (AMP), Methicillin (MET), Vancomycin (VAN), Penicillin G (PEN), Chloramphenicol (CHL), Tetracycline (TET), Gentamicin (GEN), Trimethoprim/ sulfamethoxazole (SXT), Erythromycin (ERY), Ciprofloxacin (CIP), and Levofloxacin (LVX).

Among CPS, resistance to PEN (57.9%) represents the highest resistance rate to ß-lactams, followed by AMP, FOX, MET and VAN, (26.3% each). In CPS, the only non-ß-lactam resistance was to ERY (21.1%) and TET (5.3). In the CNS, PEN (93.6%) was also the most prevalent ß-lactam resistance, followed by FOX and MET (42.6% each) and VAN (21.3%) (Table 4). Twenty staphylococcal isolates were methicillin resistant and eleven of these (55%) belonged to *S. saprophyticus* (Figure 1). The CNS showed a high resistance rate to the non-ß-lactams ERY (38.3%), followed by STX (8.5%) and TET (6.4%).

Twenty-eight (59.6%) CNS isolates and five (26.3%) CPS isolates were multidrug-resistant (Table S2). On the other hand, 6.4% of the CNS isolates and 21.1% of the CPS isolates showed no resistance to any of the antibiotics tested.

*3.3. Antimicrobial-Resistance Genes*

Among the 66 CPS and CNS isolates, the *blaZ* gene encoding penicillin resistance was the most frequently detected resistance gene (74.2%). However, among the CNS group, this percentage (87.2%) almost doubled the corresponding percentage in the CPS isolates (42.1%) (Table 5). The *mecA* gene encoding methicillin resistance was the second most frequently detected (43.9%) in all isolates (Table 5). This percentage was also higher in CNS isolates (46.8%) than in CPS (36.8%) (Table 5). The *mecA* gene was detected in all cefoxitin-resistant isolates and in 6.1% of non-cefoxitin-resistant isolates. Likewise, the co-occurrence of *mecA* and *blaZ* was detected in CPS (26.3%) and CNS (29.4%) isolates.

**Table 5.** Percentage of antibiotic-resistance genes detected in coagulase-positive (CPS) and coagulase-negative (CNS) *Staphylococcus* isolates.

| Antibiotic-Resistance Gene | CPS (*n* = 19) | CNS (*n* = 47) | Total (*n* = 66) |
|---|---|---|---|
| *bla-Z* | 8 (42.1%) | 41 (87.2%) | 49 (74.2%) |
| *mecA* | 7 (36.8%) | 22 (46.8%) | 29 (43.9%) |
| *vancA* | 6 (31.6%) | 22 (46.8%) | 28 (42.4%) |
| *vancB* | 0 | 7 (14.9%) | 7 (10.6%) |
| *ermA* | 4 (21.1%) | 3 (6.4%) | 7 (10.6%) |
| *ermB* | 1 | 7 (14.9%) | 8 (12.1%) |
| *ermC* | 5 (26.3%) | 2 (4.3%) | 7 (10.6%) |

To investigate the presence of vancomycin-resistance genes, *vancA* and *vancB* were used (Table 5). The results obtained showed that 42% of the isolates showed the *vancA* gene (26.3% among CPS and 46.8% among CNS). However, the *vancB* gene was detected only in the CNS (14.9%) (Table 5). In this group, one of the isolates that tested positive for *vancB* also tested positive for *vancA*. Three target genes were used to search for erythromycin resistance, *ermA*, *ermB* and *ermC*. The presence of erythromycin resistance genes was detected in 42.1% of the CPS isolates and in 18.2% of the CNS isolates. The *ermC* gene was the most frequent in CPS isolates (26.3%), while the *ermB* gene was the most frequent in CNS isolates (14.9%) (Table 5). Four CPS isolates (15.8%) showed the presence of *ermA* (Table 5).

**4. Discussion**

Staphylococcal enterotoxins are one of the most important causes of food poisoning. For this reason, estimating the prevalence of their coding genes in *Staphylococcus* isolates recovered from food enables risk assessment for food poisoning [52]. In this study, 32 (48.5%) of the 66 *Staphylococcus* spp. isolates analyzed harbored genes encoding enterotoxins (Figure 1). In fact, the occurrence of these genes was even similar in the CNS (33.3%) and in the CPS (31.6%). This high prevalence of SE in the CNS is very relevant and contrasts with several studies that pointed out CPS as the main source of SE in food [53]. Although the presence of enterotoxins in CNS strains has been described by other authors [53,54], few studies have been conducted to assess the presence of these enterotoxins in foodborne CNS. In one of these few studies, Moura et al. (2012) [53] reported that the occurrence of SEs in black pudding isolates was even higher in the CNS (67.5%) compared to CPS isolates (32.5%). Previous studies list the staphylococcal enterotoxins *sea*, *seb* and *sec* as the most frequent in foodborne outbreaks [54]. *Sec* and *sed* are often associated with contamination by animals, and *sea* and *seb* are associated with human contamination, through food handlers [35]. The higher prevalence of the *sec* gene (15.8% in CPS and 36.2% in CNS) in this study, followed by *sed* (12.8% in the CNS), suggests that the main source of

contamination of these RTE food was of animal origin. In fact, these RTE foods are prepared and/or sold in Maputo markets where fish and raw meat are simultaneously sold, as well as living animals, such as chickens and goats [32,33]. So, taking all results together, we may speculate that these *sec-* and *sed*-positive *Staphylococcus* spp. isolates are likely associated with cross-contamination, either indirectly through mechanical vectors or via the food matrix of animal origin [31].

The *hlb* gene encoding hemolysin was detected in nine CPS isolates (47.4%) and in seven (14.9%) CNS isolates. In CPS, 50% of the isolates that tested positive for *hlb* also tested positive for some SE-encoding genes. In the CNS, this percentage was 75%. Isolates simultaneously harboring SE- and hemolysin-encoding genes can potentially be considered more pathogenic than those positive for the SE genes but negative for *hlb*, since hemolysin holds a transversal role in pathogenicity, immune system evasion, and often in the sequestration of nutrients transported by erythrocytes [55].

Five of the twelve CPS that presented virulence genes were also resistant to antibiotics. Of these five, two were multidrug-resistant. In CNS, all the 57.4% isolates with virulence genes were antibiotic-resistant. In Mozambique, as in many other sub-Saharan African countries, there is lack of antibiotic-resistance data for *Staphylococcus* originating from food [56,57]. The results obtained in the present study show a high prevalence of resistance to β-lactams, mainly to penicillin (83.3%), methicillin (37.9%) and vancomycin (27.7%). Similar results regarding penicillin resistance (76%) were obtained for *S. aureus* isolates from raw milk in Maputo province [58]. Another study also carried out in Maputo [56] reported 89% of penicillin resistance among *S. aureus* from hospital origin. These authors suggested a selective pressure towards the emergence of resistance due to the overuse of this antibiotic. In Poland, Pyzik et al. (2019) [21] found a high resistance to penicillin G (54.3%), amoxicillin (49.6) and methicillin (16.5%) in CNS isolated from poultry. In Greece, Sergelidis et al. (2014) [59] evaluated methicillin resistance in *Staphylococcus* spp. recovered from ready-to-eat fish products and found a high resistance to penicillin and ampicillin (85.7%), erythromycin and clindamycin (71.4%), amoxicillin/clavulanic acid, gentamicin and tobramycin (57.1%) and oxacillin (28.5%) in *S. aureus*.

Among the penicillin-resistant isolates (83.3%), the *blaZ* gene was detected in 8 CPS and 41 CNS. A low percentage of methicillin resistance was found in CPS isolates (26.3%). Nhatsave et al. (2021) [58] did not find methicillin resistance in *S. aureus* isolates from raw milk in Maputo. Similar results have been reported in several southern African countries [60–62]. In South Africa, Govender et al. (2019) [63] reported 21% methicillin resistance in *S. aureus* isolated from poultry meat products, and Oosthuysen et al. (2014) [60] reported 15.3% of MRSA among *S. aureus* isolates of hospital origin. In Tanzania [61] and Uganda [62], a low prevalence of MRSA (5.4–5.7%) in hospital-origin *S. aureus* was also reported. In contrast, a high frequency of penicillin (88.2%) and methicillin (57.9%) resistance was found in CNS isolates. These results corroborate other studies that consider the CNS as a reservoir of antimicrobial-resistance genes [64–67]. In South Africa, Asante et al. [67] also reported a high occurrence of resistance to penicillin (100%) and methicillin (76.4%) in CNS isolates. In Greece, Sergelidis et al. (2014) [59] reported high prevalence of methicillin resistance (41.7%) in CNS isolated from RTE foods. In Poland, 27.6% of CNS isolated from poultry were methicillin-resistant [21]. These antibiotic-resistant CNS strains may constitute a major cause of infectious diseases, because of their potential ability to form biofilms and thus colonize community or hospital settings [64]. In addition to penicillin and methicillin resistance, a high percentage of vancomycin resistance was observed in 26.3% of the CPS isolates and in 21.3% of the CNS isolates. Among these vancomycin-resistant isolates, 33.3% of CPS and 100% of CNS were also methicillin-resistant. This vancomycin- and methicillin-resistance profile has also been reported in South Africa [67], where 46.2% of CNS isolates that were vancomycin-resistant were also methicillin-resistant. All CPS isolates exhibiting vancomycin resistance were also positive for the *vancA* gene. However, the percentage of CNS isolates with *vancA* was more than double the percentage of vancomycin-resistant isolates, suggesting the existence of non-expressed resistance genes (genotypic resistance).

While in Egypt, Al-Amery [45] reported 27% to 54% vancomycin resistance in CPS isolates from camel meat and slaughterhouse workers' hands, to our knowledge, in Mozambique, this is the first study to report the occurrence of vancomycin-resistant Staphylococci strains. According to Wu et al. [68], the prevalence of vancomycin-resistant staphylococcal strains has been increasing worldwide, from 2% in 2005 to a prevalence of 7% in 2020. In Africa, an estimated 16% of *S. aureus* isolates are vancomycin-resistant [68].

A high percentage of multidrug resistance was detected in this study (26.3% in CPS and 59.6% in CNS). In CPS, apart from the macrolide ERY, with a high percentage of resistant isolates, the relatively low-multidrug-resistance phenotype can be explained by the high susceptibility to non-ß-lactam antibiotics. Different results were reported by Mandomando et al. [69] with clinical isolates: 43% of multidrug-resistant CPS and 5% of multidrug-resistant CNS. On the other hand, Asante et al. (2021) [67] found 74.6% resistance to erythromycin, while Nhatsave et al. (2021) [58] reported a low occurrence (3%) of erythromycin resistance in *S. aureus* from raw milk in the province of Maputo. However, the results of Mandomando et al. (2010) [69] and Asante et al. (2021) [67] may be influenced by the clinical origin of the isolates. Normally, isolates of hospital origin are more resistant than those of environmental origin due to regular exposure to various antibiotics [70,71].

**5. Conclusions**

The isolates analyzed in this study were recovered from street foods with a high burden of *Staphylococcus* (potentially hazardous). The *sec* and *sed* coding genes were the most frequent virulence genes. Since *sec* and *sed* are associated with contamination of animal origin, these results highlight the need to control products of animal origin, especially at the level of pre- and post-slaughter inspection of animals. In both CPS and CNS isolates, a high percentage of penicillin resistance was observed. However, the CNS isolates showed an even higher percentage of methicillin resistance than the CPS. These results highlight the importance of CNS in parallel with CPS isolates, as both can be potentially dangerous when it comes to the possibility of causing food poisoning. In addition, CNS isolates can potentially spread antibiotic-resistance genes. To the best of our knowledge, this is the first report of vancomycin-resistant staphylococcal strains in Mozambique. These data are extremely important given the current increase in cases of vancomycin resistance, worldwide and in Africa in particular.

**Supplementary Materials:** The following supporting information can be downloaded at: https://www.mdpi.com/article/10.3390/applmicrobiol2020028/s1, Table S1: Reference 16S rRNA sequences' accession numbers from NCBI RefSeq RNA. Table S2: Prevalence of multidrug resistance in in coagulase positive (CPS) and coagulase negative (CNS) *Staphylococcus* isolates.

**Author Contributions:** Conceptualization, A.S., M.M.-F. and L.B.; methodology, A.S., M.M.-F. and L.B.; software, B.R. and M.V.C.; validation, M.M-F., L.B. and M.V.C.; formal analysis, A.S. and B.R.; investigation, A.S., J.O., M.C.; resources, M.M.-F. and L.B.; data curation, M.V.C.; writing—original draft preparation, A.S., M.M.-F., L.B.; writing—review and editing, A.S., M.M.-F., L.B., M.V.C.; supervision, L.B. and M.M.-F. All authors have read and agreed to the published version of the manuscript.

**Funding:** This work was supported by national funds through FCT—Foundation for Science and Technology, I.P., under the scholarship grants with the references PRT/BD/151521/2021 and DFA/BD/7777/2020, and institutional support of project UIDB/04129/2020 of LEAF-Linking Landscape, Environment, Agriculture and Food, Research Unit. Strategic funding from FCT to cE3c (UIDB/00329/2020), BioISI (UIDB/04046/2020) and CHANGE (LA/P/0121/2020) is also acknowledged.

**Institutional Review Board Statement:** Not applicable.

**Informed Consent Statement:** Not applicable.

**Data Availability Statement:** The authors confirm that the data supporting the findings of this study are available within the article and its Supplementary Information.

**Acknowledgments:** The authors thank Ana Carla Silva for the support with the work performed in Laboratory of Microbiology (ISA/ULisboa) and the authors thank project PORBIOTA—Portuguese E-Infrastructure for Information and Research on Biodiversity (POCI-01-0145-FEDER-022127), Operational Thematic Program for Competitiveness and Internationalization (POCI), under the PORTUGAL 2020 Partnership Agreement, through the European Regional Development Fund (FEDER).

**Conflicts of Interest:** The authors declare no conflict of interest.

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
