# Peer review of "Enterotoxin- and Antibiotic-Resistance-Encoding Genes Are Present in Both Coagulase-Positive and Coagulase-Negative Foodborne Staphylococcus Strains"

_2673-8007, doi:10.3390/applmicrobiol2020028_

Round 1

Reviewer 1 Report

Comments to the manuscript Applmicrobiol-1761405, entitled “Enterotoxin encoding genes and antibiotic resistance detected in both coagulase-positive and coagulase-negative foodborne Staphylococcus” by Acácio Salamandane, Jéssica Oliveira, Miguel Coelho, Beatriz Ramos, Mónica V. Cunha, Manuel Malfeito-Ferreira and Luísa Brito,submitted to Applied Microbiology. 

It can be said that the authors made a good comparison of the presence of virulence and antibiotic resistance genes in staphylococcal isolates from food sold on the street, pointing to the danger of coagulation of negative staphylococci as a source and spread of these genes.

The manuscript (Salamandane et al.) contains a sufficient amount of relevant data, but it was written with a significant number of errors, which indicates inattention. Also, the choice of cited references is somewhat biased, ie limited to some local authors, indicating the avoidance of citing original research, but second, third or later users. The conclusions are generally good, but we need to suggest some stronger solutions in solving the problem, and not just state the problem.

Minor/specific comments

Title: I suggest little change to be more precise, Enterotoxin and antibiotic resistance encoding genes are present in both coagulase-positive and coagulase-negative foodborne Staphylococcus strains

Abstract:

Lines 18-19. However, reports of the association of SE and methicillin-resistant Staphylococcus with coagulase 18

negative staphylococci (CNS) are beginning to re-emerge.

I suggest using the term re-emerged because there was a trend to analyze the presence of these factors in both Staphylococci groups in the 1970s, and now it’s coming back again.

Hallander HO, Sanderson H. Association of methicillin resistance to production of enterotoxin B and other factors in coagulase-positive and coagulase-negative staphylococci. Acta Pathol Microbiol Scand B Microbiol Immunol. 1972;80(2):241-5. doi: 10.1111/j.1699-0463.1972.tb00154.x.

Pyzik E, Marek A, Stępień-Pyśniak D, Urban-Chmiel R, Jarosz ŁS, Jagiełło-Podębska I. Detection of Antibiotic Resistance and Classical Enterotoxin Genes in Coagulase -negative Staphylococci Isolated from Poultry in Poland. J Vet Res. 2019;63(2):183-190. doi: 10.2478/jvetres-2019-0023. 

Lee G, Yang S. Profiles of coagulase-positive and -negative staphylococci in retail pork: prevalence, antimicrobial resistance, enterotoxigenicity, and virulence factors

Anim Biosci 2021;34(4):734-742. 
DOI: https://doi.org/10.5713/ajas.20.0660

Lines 24-26. Presence of genes encoding resistance to penicillin, methicillin, vancomycin, and erythromycin (blaZ, mecA, vancA, vancB, ermA, ermB and ermC) were also screened by PCR.

Lines 27-28. In CPS isolates, the most frequent gene was hlb (62.5%), followed by sec and sed genes (12.5%). Without respectively. 

In addition, in Table 3. Percentage of CPS isolates carrying hlb gene is 9 (47.4%), and sec sec 3 (15.8%), while sed was present in only 1 isolate. Sea and seb were not mentioned although they are also present in 1 isolate each. Please explain what is correct, why there is difference?  

Lines 29-30. Five of the 12 CPS isolates (41,7%) in which the virulence genes were detected were also resistant to at least one of the tested antibiotics.

According to Table 3. 15 CPS isolates carrying virulence genes? Not 16 as stated in Table 3. And not 12 as stated here. 

Lines 30-31. All the CNS isolates harbouring virulence genes (n=27, 57.4%) showed some of the antibiotic resistance.

According to Table 3. 41 of CNS isolates carrying virulence genes, not 27; in the case when endotoxin genes are excluded, then we have 26 isolates, but not 27. Please correct the number of strains carrying corresponding genes!!!

Lines 32-36. Regarding the presence of antibiotic resistance genes, blaZ (penicillin resistance) was the most frequent in both CPS (42.1%) and CNS (82.4%), followed by the mecA (encoding methicillin resistance) and vancA genes (vancomycin resistance), which represented 36.8% and 26.3% in CPS isolates, and 43.1% in CNS isolates, respectively.

According to Table 5. % of CNS isolates carrying blaZ gene is 87.2%, not 82.4%. In addition, mistake was detected for presence for all other resistance genes (there is no match of any of the numbers in Table 5 with those listed here).

Lines 45-46. Staphylococcus species are frequently found as commensals on human skin and nasal mucosa, but can also be found in soil, water, and food products, among other [1,2]

References (1-2) apply only to S. aureus in food and on money, while other fields far more important are not listed.

Coates R, Moran J, Horsburgh MJ. Staphylococci: colonizers and pathogens of human skin. Future Microbiol. 2014;9(1):75-91. doi: 10.2217/fmb.13.145.

Begovic J, Jovcic B, Papic-Obradovic M, Veljovic K, Lukic J, Kojic M, Topisirovic L. Genotypic diversity and virulent factors of Staphylococcus epidermidis isolated from human breast milk. Microbiol. Res 2013; 168: 77-83. doi: 10.1016/j.micres.2012.09.004

Lines 48-50. Additionally, cross-contamination from utensils and equipment, as well as from other contaminated products, particularly those subjected to handling [35], are also relevant transmission routes. 

Reference 5 has nothing to do with Staphylococcus and I do not understand why it is cited here.

Lines 50-52. The pathogenicity of S. aureus is related to the presence of several virulence factors, including S. aureus enterotoxins (SE), hemolysins encoded by hla and hlb genes and staphylokinase encoded by the sak gene [4,6].

For pathogenicity 

Foster, T. J. (2005). Immune evasion by staphylococci. Nat. Rev. Microbiol. 3(12), 948–958. doi: 10.1038/nrmicro1289

And for staphylokinase will be suitable to cite following manuscripts: 

Bokarewa MI, Jin T, Tarkowski A. Staphylococcus aureus: staphylokinase. Int J Biochem Cell Biol. 2006;38(4):504–509. doi: 10.1016/j.biocel.2005.07.005.

Line 61. These proteins are resistant to proteolysis and are thermostable [4,10].

Pleas cite manuscripts that directly analyse thermostability of Staphylococcal enterotoxins like 

Regenthal P, Hansen JS, André I, Lindkvist-Petersson K. Thermal stability and structural changes in bacterial toxins responsible for food poisoning. PLoS One. 2017;12(2):e0172445. doi: 10.1371/journal.pone.0172445. 

Tsutsuura S, Murata M. [Temperature dependence of staphylococcal enterotoxin A production by Staphylococcus aureus]. Nihon Rinsho. 2012 Aug;70(8):1323-8.

Necidová L, Bursová S,  Haruštiaková D, Bogdanovičová K, I. Lačanin I. Effect of heat treatment on activity of staphylococcalenterotoxins of type A, B, and C in milk.  J. Dairy Sci. 102:3924–3932 https://doi.org/10.3168/jds.2018-15255

Lines 68-69. Antibiotic resistance, particularly methicillin-resistant Staphylococcus aureus (MRSA), 68

is another problem associated with staphylococci [4,9]. Please cite reference that explain mechanism of methicillin resistance like:

Stapleton PD, Taylor PW. Methicillin resistance in Staphylococcus aureus: mechanisms and modulation. Sci Prog. 2002;85(Pt 1):57-72. doi: 10.3184/003685002783238870. 

Lines 78-80. When associated with poor hygiene practices by handlers and poor hygienic conditions of the exposure environment, ready-to-eat (RTE) street food present a potential risk of transmission of staphylococcal strains [2124]. Regarding hygiene and street food authors cited only manuscripts of one laboratory (5), but this problem is common for many countries? There are so many manuscripts dealing with this theme like:

Sivakumar M, Dubal ZB, Kumar A, Bhilegaonkar K, Vinodh Kumar OR, Kumar S, Kadwalia A, Shagufta B, Grace MR, Ramees TP, Dwivedi A. Virulent methicillin resistant Staphylococcus aureus (MRSA) in street vended foods. J Food Sci Technol. 2019;56(3):1116-1126. doi: 10.1007/s13197-019-03572-5. 

Kadariya J, Smith TC, Thapaliya D. Staphylococcus aureus and staphylococcal food-borne disease: an ongoing challenge in public health. Biomed Res Int. 2014;2014:827965. doi: 10.1155/2014/827965. 

Line 87. This study used 70 isolates of presumptive Staphylococcus previously recovered……

Also, later authors use term “presumptive Staphylococcus”. For me they (isolates) should be determined as Staphylococcus.

Lines 92-93. From each plate that presented well isolated colonies and characteristics of Staphylococcus, a CFU was selected.

I do not understand this sentence. What is mean CFU was selected/ maybe determined?

Line 103. …..16S rRNA gene (1500 bp) with Bac27F forward (5-AGAGTTTGGATCMTGGCTCAG-3) and Univ1492R universal reverse (5-CGGTTACCTTGTTACGACTT-3) primers [5,28].

Here is not necessary to cite these manuscripts because they are not original that proposed these primers for sequencing and in addition these primers are universal. 

Lines 132-133. Table 1. Primers used in this study for the detection of virulence genes, size of expected amplified product and annealing temperature (adapted from Adame-Gómez et al [2]).

It is a bit problematic to just take a part of the Table from a manuscript where the authors proposed only one pair of new staphylokinase primers, while for the others they cited works in which they were proposed for the first time.

Lines 182-183.  The 70 presumptive staphylococcal isolates used in this study were identified based on 16S rRNA gene sequence analysis (Figure 1).

Lines 184-185. The most representative CNS species are S. warneri (22.9%), S. saprophyticus (18.6%), S. xylosus (10%) and S. pasteuri (5.6%).

In total it is 84,1%, What is rest should be stated?

Virulence genes were detected among the 66 identified Staphylococcus isolates

66 isolates of 70 tested represent 94.3%. Please be more precise? 

Figure 1 is very low resolution and difficult to read data from. Increase image resolution.

Line 269. In one of these studies, Moura et al [40] reported that the occurrence…..

Too many "these" are repeated in this paragraph, so it is difficult to follow what refers to CNS strains or enterotoxies, although the authors did not analyze the presence of enterotoxins but genes encoding enterotoxins.

Most of the works referred to in the Discussion come from the African continent or Asia, it would be good to compare in more detail with the findings from other continents, such as America, which also has a share of street food sales.

Author Response

Comments to the manuscript Applmicrobiol-1761405, entitled “Enterotoxin encoding genes and antibiotic resistance detected in both coagulase-positive and coagulase-negative foodborne Staphylococcus” by Acácio Salamandane, Jéssica Oliveira, Miguel Coelho, Beatriz Ramos, Mónica V. Cunha, Manuel Malfeito-Ferreira and Luísa Brito,submitted to Applied Microbiology. 

It can be said that the authors made a good comparison of the presence of virulence and antibiotic resistance genes in staphylococcal isolates from food sold on the street, pointing to the danger of coagulation of negative staphylococci as a source and spread of these genes.

The manuscript (Salamandane et al.) contains a sufficient amount of relevant data, but it was written with a significant number of errors, which indicates inattention. Also, the choice of cited references is somewhat biased, ie limited to some local authors, indicating the avoidance of citing original research, but second, third or later users. The conclusions are generally good, but we need to suggest some stronger solutions in solving the problem, and not just state the problem.

Answer: Dear reviewer, thank you for all your comments and contributions to the improvement of our manuscript. Some of the answers to your comments were posted in the revised manuscript and highlighted in track change.

Minor/specific comments

  1. Title: I suggest little change to be more precise, Enterotoxin and antibiotic resistance encoding genes are present in both coagulase-positive and coagulase-negative foodborne Staphylococcus strains

Answer: Done, according to your suggestion

Abstract:

  1. Lines 18-19. However, reports of the association of SE and methicillin-resistant Staphylococcus with coagulase

negative staphylococci (CNS) are beginning to re-emerge.

I suggest using the term re-emerged because there was a trend to analyze the presence of these factors in both Staphylococci groups in the 1970s, and now it’s coming back again.

Answer: Thank you for suggested references: Done, according to suggestion, References number [20, 21, 22].

Hallander HO, Sanderson H. Association of methicillin resistance to production of enterotoxin B and other factors in coagulase-positive and coagulase-negative staphylococci. Acta Pathol Microbiol Scand B Microbiol Immunol. 1972;80(2):241-5. doi: 10.1111/j.1699-0463.1972.tb00154.x.

Pyzik E, Marek A, Stępień-Pyśniak D, Urban-Chmiel R, Jarosz ŁS, Jagiełło-Podębska I. Detection of Antibiotic Resistance and Classical Enterotoxin Genes in Coagulase -negative Staphylococci Isolated from Poultry in Poland. J Vet Res. 2019;63(2):183-190. doi: 10.2478/jvetres-2019-0023. 

Lee G, Yang S. Profiles of coagulase-positive and -negative staphylococci in retail pork: prevalence, antimicrobial resistance, enterotoxigenicity, and virulence factors

Anim Biosci 2021;34(4):734-742. 
DOI: https://doi.org/10.5713/ajas.20.0660

  1. Lines 24-26. Presence of genes encoding resistance to penicillin, methicillin, vancomycin, and erythromycin (blaZ, mecA, vancA, vancB, ermA, ermB and ermC) were also screened by PCR.

Answer: Done. Line 25

  1. Lines 27-28. In CPS isolates, the most frequent gene was hlb (62.5%), followed by sec and sed genes (12.5%). Without respectively.

Answer: Corrected. Line 28-31 

In addition, in Table 3. Percentage of CPS isolates carrying hlb gene is 9 (47.4%), and sec sec 3 (15.8%), while sed was present in only 1 isolate. Sea and seb were not mentioned although they are also present in 1 isolate each. Please explain what is correct, why there is difference? 

Answer: we apologize for the confusion. Corrected.

  1. Lines 29-30. Five of the 12 CPS isolates (41,7%) in which the virulence genes were detected were also resistant to at least one of the tested antibiotics.

According to Table 3. 15 CPS isolates carrying virulence genes? Not 16 as stated in Table 3. And not 12 as stated here.

Answer: Effectively, there were 12 isolates with virulence genes (figure 1). However, in four isolates coexistence of at least two genes was observed.

  1. Lines 30-31. All the CNS isolates harbouring virulence genes (n=27, 57.4%) showed some of the antibiotic resistance.

According to Table 3. 41 of CNS isolates carrying virulence genes, not 27; in the case when endotoxin genes are excluded, then we have 26 isolates, but not 27. Please correct the number of strains carrying corresponding genes!!!

Answer: There are effectively 27 isolates with virulence genes. The number 41 results from co-existence of more than one virulence gene in 41 from the 27 CNS isolates.

  1. Lines 32-36. Regarding the presence of antibiotic resistance genes, blaZ (penicillin resistance) was the most frequent in both CPS (42.1%) and CNS (82.4%), followed by the mecA (encoding methicillin resistance) and vancA genes (vancomycin resistance), which represented 36.8% and 26.3% in CPS isolates, and 43.1% in CNS isolates, respectively.

According to Table 5. % of CNS isolates carrying blaZ gene is 87.2%, not 82.4%. In addition, mistake was detected for presence for all other resistance genes (there is no match of any of the numbers in Table 5 with those listed here).

Answer: Thank you very much for your attention. Due to a small change in the total number of species confirmed as staphylococci, instead of 70 they are 66, we had to update the results in the tables and, by mistake, this update was not done in the abstract. Correction made.

  1. Lines 45-46. Staphylococcus species are frequently found as commensals on human skin and nasal mucosa, but can also be found in soil, water, and food products, among other [1,2]

References (1-2) apply only to S. aureus in food and on money, while other fields far more important are not listed.

Coates R, Moran J, Horsburgh MJ. Staphylococci: colonizers and pathogens of human skin. Future Microbiol. 2014;9(1):75-91. doi: 10.2217/fmb.13.145.

Begovic J, Jovcic B, Papic-Obradovic M, Veljovic K, Lukic J, Kojic M, Topisirovic L. Genotypic diversity and virulent factors of Staphylococcus epidermidis isolated from human breast milk. Microbiol. Res 2013; 168: 77-83. doi: 10.1016/j.micres.2012.09.004

Answer: Thank you for suggested references. They were added in the text. References number [3,4].

  1. Lines 48-50. Additionally, cross-contamination from utensils and equipment, as well as from other contaminated products, particularly those subjected to handling [3–5], are also relevant transmission routes. 

Reference 5 has nothing to do with Staphylococcus and I do not understand why it is cited here.

Answer: Removed

Lines 50-52. The pathogenicity of S. aureus is related to the presence of several virulence factors, including S. aureus enterotoxins (SE), hemolysins encoded by hla and hlb genes and staphylokinase encoded by the sak gene [4,6].

For pathogenicity 

Foster, T. J. (2005). Immune evasion by staphylococci. Nat. Rev. Microbiol. 3(12), 948–958. doi: 10.1038/nrmicro1289

And for staphylokinase will be suitable to cite following manuscripts: 

Bokarewa MI, Jin T, Tarkowski A. Staphylococcus aureus: staphylokinase. Int J Biochem Cell Biol. 2006;38(4):504–509. doi: 10.1016/j.biocel.2005.07.005.

 Answer:  Thank you for suggested references. Done, according to suggestion, References number [8,9].

  1. Line 61. These proteins are resistant to proteolysis and are thermostable [4,10].

Pleas cite manuscripts that directly analyse thermostability of Staphylococcal enterotoxins like 

Regenthal P, Hansen JS, André I, Lindkvist-Petersson K. Thermal stability and structural changes in bacterial toxins responsible for food poisoning. PLoS One. 2017;12(2):e0172445. doi: 10.1371/journal.pone.0172445. 

Tsutsuura S, Murata M. [Temperature dependence of staphylococcal enterotoxin A production by Staphylococcus aureus]. Nihon Rinsho. 2012 Aug;70(8):1323-8.

Necidová L, Bursová S,  Haruštiaková D, Bogdanovičová K, I. Lačanin I. Effect of heat treatment on activity of staphylococcalenterotoxins of type A, B, and C in milk.  J. Dairy Sci. 102:3924–3932 https://doi.org/10.3168/jds.2018-15255

Answer:  Thank you for suggested references. Done, according to suggestion, References number [14,15].

  1. Lines 68-69. Antibiotic resistance, particularly methicillin-resistant Staphylococcus aureus (MRSA), 68

is another problem associated with staphylococci [4,9]. Please cite reference that explain mechanism of methicillin resistance like:

Stapleton PD, Taylor PW. Methicillin resistance in Staphylococcus aureus: mechanisms and modulation. Sci Prog. 2002;85(Pt 1):57-72. doi: 10.3184/003685002783238870. 

Answer:  Thank you for suggested reference. Done, according to suggestion, Reference number [23].

  1. Lines 78-80. When associated with poor hygiene practices by handlers and poor hygienic conditions of the exposure environment, ready-to-eat (RTE) street food present a potential risk of transmission of staphylococcal strains [21–24]. Regarding hygiene and street food authors cited only manuscripts of one laboratory (5), but this problem is common for many countries? There are so many manuscripts dealing with this theme like:

Sivakumar M, Dubal ZB, Kumar A, Bhilegaonkar K, Vinodh Kumar OR, Kumar S, Kadwalia A, Shagufta B, Grace MR, Ramees TP, Dwivedi A. Virulent methicillin resistant Staphylococcus aureus (MRSA) in street vended foods. J Food Sci Technol. 2019;56(3):1116-1126. doi: 10.1007/s13197-019-03572-5. 

Kadariya J, Smith TC, Thapaliya D. Staphylococcus aureus and staphylococcal food-borne disease: an ongoing challenge in public health. Biomed Res Int. 2014;2014:827965. doi: 10.1155/2014/827965.

Answer: Thank you for observation. Other reference as now added.   Reference number [35,36].

  1. Line 87. This study used 70 isolates of presumptive Staphylococcus previously recovered……

Also, later authors use term “presumptive Staphylococcus”. For me they (isolates) should be determined as Staphylococcus.

Answer:  Before molecular identification we classified all isolates by biochemical tests and so they were 70 isolates (19 coagulase positive and 51 coagulase negative). After molecular identification, four isolates were allocated to another genus, so we chose to designate in the methodology the 70 as presumptive Staphylococcus.

  1. Lines 92-93. From each plate that presented well isolated colonies and characteristics of Staphylococcus, a CFU was selected. I do not understand this sentence. What is mean CFU was selected/ maybe determined?

Answer: The sentence has been rewritten to clarify.

  1. Line 103. …..16S rRNA gene (1500 bp) with Bac27F forward (5-AGAGTTTGGATCMTGGCTCAG-3) and Univ1492R universal reverse (5-CGGTTACCTTGTTACGACTT-3) primers [5,28].

Here is not necessary to cite these manuscripts because they are not original that proposed these primers for sequencing and in addition these primers are universal.

Answer: Done. References removed

  1. Lines 132-133. Table 1. Primers used in this study for the detection of virulence genes, size of expected amplified product and annealing temperature (adapted from Adame-Gómez et al [2]).

It is a bit problematic to just take a part of the Table from a manuscript where the authors proposed only one pair of new staphylokinase primers, while for the others they cited works in which they were proposed for the first time.

Answer: Thank you for you observation. The original reference has been cited.

  1. Lines 182-183.  The 70 presumptive staphylococcal isolates used in this study were identified based on 16S rRNA gene sequence analysis (Figure 1).

Answer: Done. According to the suggestion

  1. Lines 184-185. The most representative CNS species are S. warneri (22.9%), S. saprophyticus (18.6%), S. xylosus (10%) and S. pasteuri (5.6%).

In total it is 84,1%, What is rest should be stated?

Virulence genes were detected among the 66 identified Staphylococcus isolates

66 isolates of 70 tested represent 94.3%. Please be more precise?

Answer: Corrected

  1. Figure 1 is very low resolution and difficult to read data from. Increase image resolution.

Answer: New image has been added

  1. Line 269. In one of these studies, Moura et al [40] reported that the occurrence…..

Too many "these" are repeated in this paragraph, so it is difficult to follow what refers to CNS strains or enterotoxies, although the authors did not analyze the presence of enterotoxins but genes encoding enterotoxins.

Most of the works referred to in the Discussion come from the African continent or Asia, it would be good to compare in more detail with the findings from other continents, such as America, which also has a share of street food sales.

Answer: Unfortunately, RTE street food is characteristic of developing countries like most African and Asian countries. Eventually, RTE street food sales occurs in some Latin American countries, which can be compared with RTE street food in Africa and Asia. However, specifically for virulence genes and antibiotic resistance profile in RTE street food staphylococci, we did not find many studies in countries other than Africa and Asia.

Nevertheless, discussion has been improved and some studies from other regions have been added [21,59]

Reviewer 2 Report

The manuscript Enterotoxin encoding genes and antibiotic resistance detected in both coagulase-positive and coagulase-negative foodborne Staphylococcus is very consistent, structured and well defined. The introduction is informative and consistent, the methods are well described, and the results are well presented. The manuscript has potential for publication consideration, but some issues must be solved before that:

-Introduction: The authors should explain that there was broader research, but they just focused on Staphylococcus.

-line 166: the authors should explain in more detail how did they achieve concertation of cells 10^6 CFU/mL.

-line 167: The authors should add information on the volume of added bacteria on solid media.

-line 173: The authors should add information on repetition of the test.

-Figure 1: increase the resolution of the figure, it is hard to read.

Author Response

The manuscript Enterotoxin encoding genes and antibiotic resistance detected in both coagulase-positive and coagulase-negative foodborne Staphylococcus is very consistent, structured and well defined. The introduction is informative and consistent, the methods are well described, and the results are well presented. The manuscript has potential for publication consideration, but some issues must be solved before that:

Answer: Dear reviewer, thank you for all your comments and contributions to the improvement of our manuscript. The highlighted of the answers were posted in track change.

  1. Introduction: The authors should explain that there was broader research, but they just focused on Staphylococcus.

Answer: Done

  1. line 166: the authors should explain in more detail how did they achieve concertation of cells 10^6 CFU/mL.

Answer: Done: line 169-170

  1. line 167: The authors should add information on the volume of added bacteria on solid media.

Answer: Done. Line 171

  1. line 173: The authors should add information on repetition of the test.

Answer: Done. Line 177-178

  1. Figure 1: increase the resolution of the figure, it is hard to read.

Answer: New image has been added

Reviewer 3 Report

Dear Authors,

The presented manuscript aimed to investigate the presence of staphylococcal enterotoxin genes and to characterize the phenotypic and genotypic antimicrobial resistance in 66 isolates of Staphylococcus spp. (47 CNS and 19 CPS) recovered from ready-to-eat (RTE) street food sold in Maputo, Mozambique.

The introduction provides a good, generalized background of the topic that quickly gives the reader appreciation of the scientific relevance and timeliness of the research theme. The study meets the criterion of originality. However the main weak points of the manuscript are lack of short characterization the source of investigated bacterial strains, lack of statistical analysis and some technical inaccuracies.

I have for Authors some suggestions for improvement as follows:

  • Line 27; 88:  Please, specify what products (RTE) were used?
  • Line 82-85: Please, add a comment that this study is a follow-up to previous research by authors (reference no. 21),
  • Line 102: after “… as previously described..” add “by Adame-Gómez et al. (2020)..”
  • Line 132: change to Adame-Gómez et al. (2020),
  • Table 1, 2 etc.”- this content  has to be bolded in the text of manuscript,
  • Line 149: change to „Prunier et al. (2003)”,
  • Line 150: change to “Al-mery et al. (2019)”. The authors' names must agree with the quotation in section References,
  •   Please, complete the Statistical Analyses section in the manuscript. This is necessary to show, for instance a statistically significant differences in terms of antibiotic resistance. Without that the presented manuscript is poor.
  • Line 269: change to “Moura et al. (2012)”,
  • The Discussion should be taken into account, a source (ready-to-eat (RTE) street food sold from Maputo, Mozambique) of investigated bacterial strains,
  • Line 341-342: please, delete the reference no 21. It is not a conclusion of presented manuscript.

From my standpoint, the presented  manuscript will be appropriated for publication Journal Applied Microbiology, after major revision, given the above aspects.

Author Response

Dear Authors,

The presented manuscript aimed to investigate the presence of staphylococcal enterotoxin genes and to characterize the phenotypic and genotypic antimicrobial resistance in 66 isolates of Staphylococcus spp. (47 CNS and 19 CPS) recovered from ready-to-eat (RTE) street food sold in Maputo, Mozambique.

The introduction provides a good, generalized background of the topic that quickly gives the reader appreciation of the scientific relevance and timeliness of the research theme. The study meets the criterion of originality. However, the main weak points of the manuscript are lack of short characterization the source of investigated bacterial strains, lack of statistical analysis and some technical inaccuracies.

Answer: Dear reviewer, thank you for all your comments and contributions to the improvement of our manuscript. Some of the answers to reviewers’ comments were posted in the revised manuscript and highlighted in track change.

I have for Authors some suggestions for improvement as follows:

  1. Line 27; 88:  Please, specify what products (RTE) were used?

Answer: Different and typical street foods from Mozambique were the source of the isolates analyzed in this work, making difficult to list them. We add now additional information as well as the reference on line 90

  1. Line 82-85: Please, add a comment that this study is a follow-up to previous research by authors (reference no. 21),

Answer: Thank you for suggestion. Done. Line 83-84

  1. Line 102: after “… as previously described..” add “by Adame-Gómez et al. (2020)..”

Answer: Done. Line 105

  1. Line 132: change to Adame-Gómez et al. (2020),

Answer: Done.

  1. Table 1, 2 etc.”- this content has to be bolded in the text of manuscript,
  2. Line 149: change to „Prunier et al. (2003)”,

Answer: Done

  1. Line 150: change to “Al-mery et al. (2019)”. The authors' names must agree with the quotation in section References,

Answer: Done. In all references

  1.   Please, complete the Statistical Analyses section in the manuscript.

Answer: In this work, we opted for a descriptive analysis since the number of CNS (47) is much higher than the number of CPS (19). Unequal sample sizes can lead to unequal variances between samples, which affects the assumption of equal variances in tests like ANOVA.

Answer:

  1. Line 269: change to “Moura et al. (2012)”,

Answer: Done.

  1. The Discussion should be taken into account, a source (ready-to-eat (RTE) street food sold from Maputo, Mozambique) of investigated bacterial strains,

Answer: Discussion has been improved

  1. Line 341-342: please, delete the reference no 21. It is not a conclusion of presented manuscript.

Answer: Done

Round 2

Reviewer 3 Report

Dear Authors,

Majority of previous comments have been taken into account.             

I haven’t more  objections to the new version of the manuscript.

From my point of view the new version of the article is appropriate for publication in Journal – Applied Microbiology ,  based on the above comment.